# Embodiment of an artificial limb in mice

Zineb Hayatou[1], Hongkai Wang[2], Antoine Chaillet[3], Daniel E. Shulz[1], Valérie Ego-Stengel[1], Luc Estebanez 🔘[1]*

1 Université Paris-Saclay, CNRS, Institut des Neurosciences Paris-Saclay (NeuroPSI), Saclay, France, 2 Faculty of Medicine, Dalian University of Technology, Dalian City, China, 3 Université Paris-Saclay, CNRS, CentraleSupélec, Laboratoire des signaux et systèmes, Gif-Sur-Yvette, France

* luc.estebanez@cnrs.fr

## Abstract

Body ownership disorders can be triggered by disease or body damage. Methods to probe limb embodiment are required to address those disorders. This includes the development of neuroprostheses that better integrate into the body scheme of the user. To this end, the "rubber hand illusion" protocol is a key behavioral method to probe the powerful embodiment that can be triggered by congruent somatosensory and visual inputs from the limb. So far, the neurophysiology of limb embodiment remains poorly known, in part because translating the rubber hand illusion to animal models such as the mouse remains challenging. Yet, mapping out the brain circuits of embodiment thanks to the use of genetic and optogenetic research tools would allow to propose novel embodiment restoration strategies. Here, we show that the rubber hand illusion described in humans can be translated to the mouse forelimb model using an automated, videography-based procedure. We exposed head-fixed mice to a visible, static 3D-printed replica of the right forelimb, while their own forelimb was hidden from their sight. We synchronously brushed their hidden forelimb and the replica. Following these visuo-tactile associations, the replica was visually threatened, and we probed the reaction of the mice using automated tracking of pupils and facial expression. The mice focused significantly more of their gaze toward the threatened forelimb replica after receiving synchronous tactile and visual information compared to asynchronous. More generally, across test and control conditions, the mouse pupillary response was consistent with the human overt response to the rubber hand illusion. Thus, our results show that mice exhibit quantifiable behavioral markers of the embodiment of an artificial forelimb.

## Introduction

We perceive our limbs as part of ourselves, and we feel threatened by any menace to them. This sense of embodiment can be disrupted by brain lesions leading to a loss of recognition of body parts, and even to their active rejection from the body

**Data availability statement:** All experimental data and analysis code is stored in the Zenodo repository DOI 10.5281/zenodo.14635566, available at the following link: https://zenodo.org/records/14635566.

**Funding:** This work was funded by LE PRC Hermin, ANR. https://anr.fr/ to LE, JCJC Mesobrain, ANR. https://anr.fr/ to LE, PRC Expect, ANR. https://anr.fr/ to DES, PRC PerBaCo, ANR. https://anr.fr/ to DES, Fondation 3DS https://www.lafondation3ds.org/fr/ to LE, 80|Prime 2020, CNRS, https://www.cnrs.fr/ to LE and AC, MITI PRIME 2024, CNRS, https://www.cnrs.fr/ to LE and AC, PRC MotorSense, ANR. https://anr.fr/ to VE, RISE iNavigate, 873178, Horizon 2020 Framework Programme. https://cordis.europa.eu/project/id/873178 to DES, OI hCODE, Université Paris-Saclay, https://www.universite-paris-saclay.fr/objets-interdis-ciplinaires/h-code to LE. The funders played no role in the design of the study, data collection and analysis, preparation of the manuscript, or decision du publish.

**Competing interests:** The authors have declared that no competing interests exist.

**Abbreviation:** SCR, skin conductance response.

representation [1]. In the case of amputated patients, efficient use of a prosthesis can be hampered by a lack of prosthesis embodiment, thereby causing a progressive decrease in daily prosthesis use and finally an abandonment of the prosthetic limb [2,3]. In contrast, prosthesis embodiment is associated with a reduction in the sensations arising from the "phantom" of the missing limb, including painful perceptions [4].

In an experimental setting, it is possible to either build or disrupt the sense of embodiment of an artificial limb by manipulating the temporal coincidence of tactile and visual stimulations. This multisensory strategy has been widely used with human participants to study the sense of body ownership and embodiment [5,6]. In particular, in the "rubber hand illusion" experiment, a rubber limb is placed in a position that makes it visible to the participant, while its real hand is hidden from view. Both the hidden real hand and the visible artificial hand are stimulated in synchrony with a brush [5]. A large fraction of the subjects of these experiments report that after stimulation, the rubber hand they are seeing is their real hand [7,8]. These verbal reports are consistent with overt behaviors of the subject during the experiment, and in particular with a fear-like reaction when the artificial hand is visibly threatened or "injured" by the experimenter [9]. This has been assessed through high skin conductance responses (SCRs) and reports of participants showing signs of anxiety or pain anticipation just before the injury of the prosthesis [9]. In addition, an increased activity was observed in the insula and anterior cingulate cortex, two regions that are associated with anxiety and interoceptive awareness [10].

By building on the ability of visuo-tactile synchronized inputs to induce embodiment, it is possible to achieve the embodiment of an artificial device, including robotic human prostheses, by stimulating the stump of amputees [11–13]. This demonstrates the flexibility of this body-object pairing mechanism, which may be a gateway toward embodied neuroprosthetics [14]. However, this does not generalize to all object shapes, as studies showed that the object needs to look like a hand [15] or share functional similarity with the hand [16] for successful embodiment. When a hand-shaped object was replaced by a non-hand-shaped object, reports showed a significantly weaker embodiment [15,17,18].

So far, the physiological bases of this sensory-based forelimb embodiment remain unclear. This is partly due to the lack of an animal model to study embodiment. To address this, embodiment experiments have been carried out in macaques and mice [19–23]. In mice, simultaneous tactile stimulations were applied to a rubber tail and to the hidden tail of the mouse, and the strength of head movements toward the tail was estimated in reaction to an experimenter visibly grasping the rubber tail at the end of the stimulations [22,23]. These studies suggest that it is possible to study embodiment in mice. However, they focused on the embodiment of a body part that is largely unseen by the animal, that is not present in humans, and using a protocol that required manual behavioral scoring.

In this study, we propose to examine the embodiment of the forelimb of mice. Indeed, mice can use their forelimb for rich and complex behaviors, including cortically dependent reaching [24], adjustment of a joystick position [25], as well as

manipulation of food with complex shapes [26]. Thus, the mouse forelimb constitutes a relevant model for the study of the upper limb function in human.

Moreover, we aimed to probe the embodiment of an artificial forelimb in the mouse by using an automated test based on the imaging and tracking of the mouse body features.

In our protocol, head-fixed mice were presented with an artificial, static replica of their right forelimb at a plausible physiological location. At the same time, their forelimb was hidden from sight and held in place below the platform where the artificial limb was located. During a two-minute pairing sequence, we applied mechanically controlled synchronized brush strokes to the real and the artificial forelimbs. This was immediately followed by the rapid drop of a sharp object toward the artificial forelimb that was within the visual field of the animal. During this sequence, we tracked the gaze of the mice with high-speed videography. Consistent with the broader literature on the rubber hand illusion, we observe a stronger behavioral response under synchronous conditions [9,10]. In our case, this is reflected by a significant gaze shift toward the incoming—and potentially threatening—object when brush strokes were synchronous (versus asynchronous) and when the artificial forelimb looked similar to an actual biological limb (versus a white cube-shaped object).

## Results

### Longer gaze in the direction of the threatened artificial limb following synchronous stimulations

We first habituated mice to head fixation. We then carried behavioral sessions during which an artificial right forelimb was positioned next to the head-fixed mouse, in a position that was within its field of view. At the same time, the own right forelimb of the mouse was hidden from sight. During these behavioral sessions, we performed high-speed imaging of the mouse face to track several features (see Materials and methods), in particular pupil positions.

During the pairing stage of the protocol (Fig 1A left), both the artificial forelimb and the physiological forelimb received brush strokes, either simultaneously (synchronous stimulation, Fig 1B left), or randomly time-shifted (asynchronous stimulation, Fig 1B right). We tracked pupil position as well as other body points during each trial using high-speed videography (Fig 1C). During baseline and brush stimulations (single trials in Fig 2A), we found no significant difference in the behavior of the mice between the synchronous and asynchronous conditions. Thus, any subsequent difference in behavior after the stimulations is unlikely to originate from an initial bias.

After the pairing with brush strokes, the artificial forelimb was rapidly approached by an arrow-like object (Threat, Fig 1A, right). In both the synchronous and asynchronous pairing conditions, the mice responded to this event with a rapid, mainly horizontal movement of the right pupil toward the artificial limb and threatening object (Fig 2A, 2B). Overall, ten mice were exposed to the protocol. Five sessions were run for each experiment; one session per day. During each session, the mice were exposed to two trials (synchronous and asynchronous pairing). The presentation order of these two trials changed in each session (S1 Fig).

One second after this first response, the mice behavior diverged between the two conditions. In the synchronous condition, on average, the mice looked again in the direction of the menace and artificial limb, while in the asynchronous condition, the mice stopped looking in this direction and moved their right pupil back to the resting position (Fig 2C–2E).

We tested for the significance of the difference in horizontal pupillary position between the two conditions. To this aim, we performed a bootstrap analysis by taking advantage of the 120 s baseline that we systematically recorded before pairing time. We estimated from the baseline distribution of ocular positions a significance threshold (see Materials and methods) that revealed two windows of interest corresponding to the two phases of the pupillary behavior described earlier (Fig 2F).

Measuring in individual mice the mean horizontal right pupil position during the two windows (W1 and W2) revealed a diversity of baseline positions (Fig 2G). Further, testing the difference between the synchronous and asynchronous condition with a Wilcoxon parametric test revealed a significant difference between the two conditions ($p = 0.014$) in the W2 window corresponding to the pupillary movement toward the threat. We then carried the same analysis, looking this time

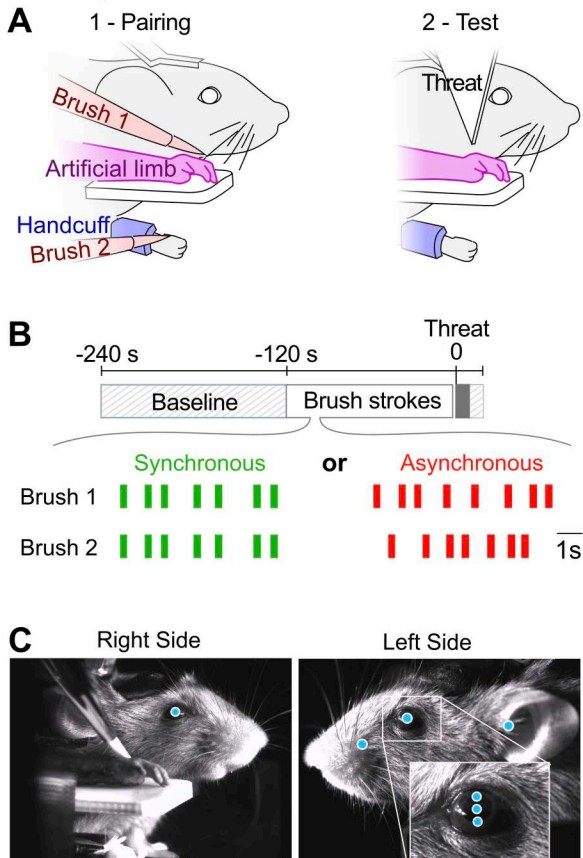

**Fig 1. A forelimb embodiment test in the mouse model. (A)** The two main phases of the embodiment test. Left: Pairing step. During 120 seconds, brush stimulations were applied with brush 1 to the artificial limb (visual input) and with brush 2 to the corresponding forelimb (tactile input), which was hidden from the mouse sight. Right: Testing step. Embodiment of the artificial limb was tested by measuring the intensity of the response of the mouse to a rapidly incoming, sharp object (labeled Threat) that targeted the artificial limb. **(B)** Time sequence of the experiment. Top: timeline of the pairing and test sections of an individual trial, starting with a 120 s baseline idle time. 120 s of brush paring, followed by the rapid incoming of the threat, which stayed fixed during 10 s. Bottom: example brush stroke times for a synchronous (green) and an asynchronous (red) trial. Line thickness scales are at scale with the 300 ms duration of the brush strokes. **(C)** Views from the right and left sides of the mouse acquired by high-speed cameras during the pairing stage. Cyan dots: points of interest that are tracked, including the pupil center position and diameter (measured via 2 points in the vertical axis, see close-up) for both eyes, as well as points on the left whisker pad and the left ear.

at the left pupil (Fig 2H). We found highly similar results, although the differences were smaller than for the right pupil. Note that in W2, the Wilcoxon test was also significant ($p = 0.02$).

Overall, in this first series of experiments, the mice moved their eye significantly more toward the threat to the artificial forelimb following synchronous pairing. These results are consistent with an embodiment of the artificial forelimb.

## Reduced reaction to the threat when the artificial limb is replaced by a cube-like shape

We then asked if mice would react similarly to the threat if the artificial limb is replaced by an object that does not resemble their forelimb. Therefore, in a second series of experiments, we exposed the same mice ($n = 9$ following the removal of one animal, see Materials and methods) to the same protocol as in Fig 2, but replacing the artificial limb with a white rectangular block (Fig 3A). In these experiments, we did find that there was a limited but significant difference in average right pupil shift in response to the threat (Fig 3B, 3C).

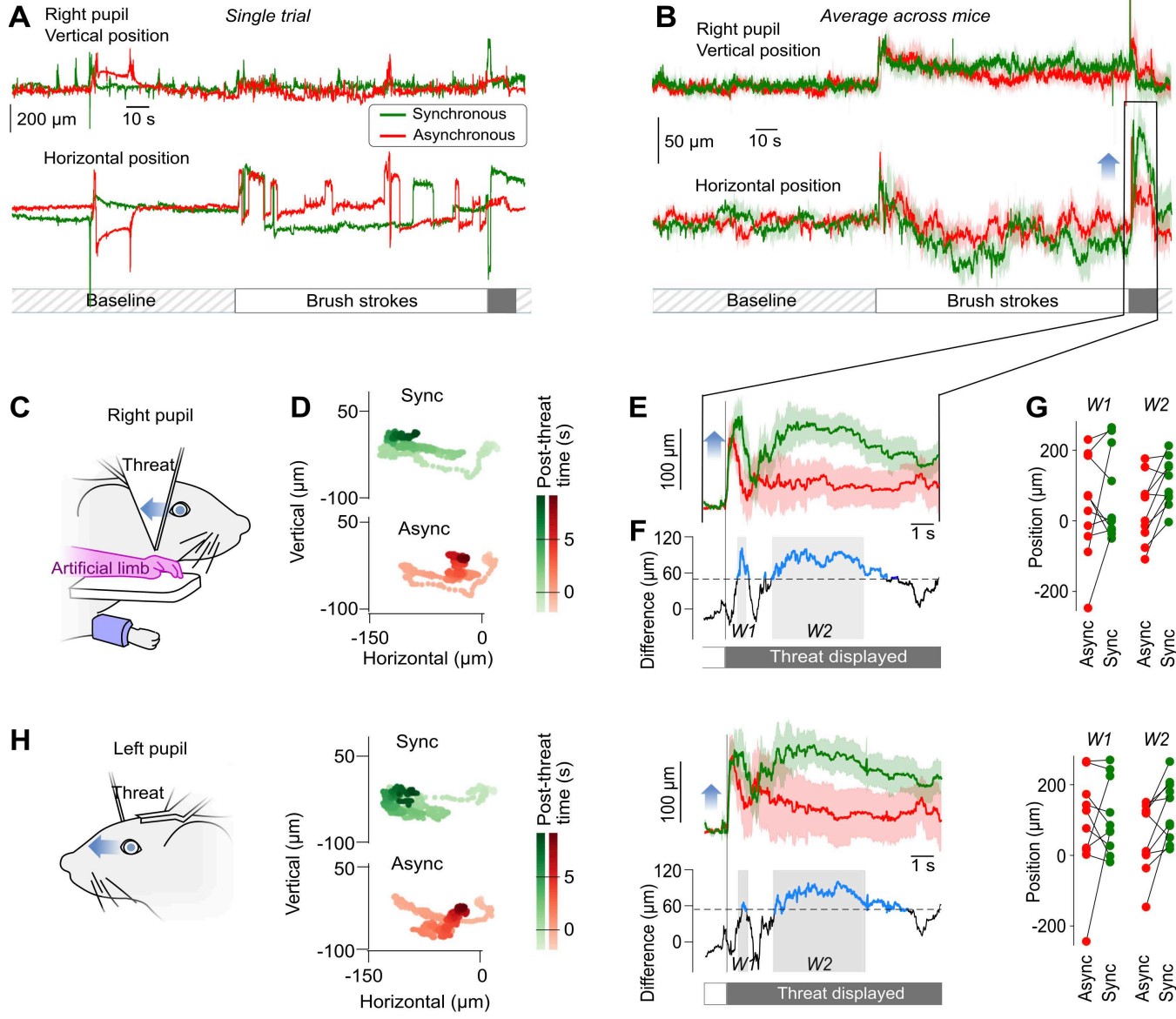

**Fig 2. Pupil shifts in the direction of the threatened artificial limb are longer after synchronous stimulation. (A)** Example of vertical (top) and horizontal (bottom) movements of the right pupil during a synchronous (green) and an asynchronous (red) trial for one session. The sequence includes a Baseline, Brush strokes pairing, and a threat to the artificial forelimb. **(B)** Average vertical (top) and horizontal (bottom) movements of the right pupil during synchronous and asynchronous condition trials, normalized to the average position during the 120 s baseline ($n = 10$ mice). **(C)** Schematic of the right side of the mouse face during the experiment. Blue arrow: general direction of the pupillary movement following the threat. **(D)** Spatial trajectory of the right pupil position from 1 s before to 7 s after the threat starts to be displayed. Top: synchronous pairing. Bottom: asynchronous pairing. **(E)** Top: Average horizontal movements of the right pupil following the threat onset, normalized relative to the average position 1 s before the threat ($n = 10$). Light background: SEM. Blue arrow: direction of pupil movement as in C. **(F)** Average difference between the right pupil movements in the two conditions in E. Blue sections: significant differences (Bootstrap based test $p < 0.05$). Black dashed line: significance threshold. Gray background: W1 and W2 time windows selected for further quantification. **(G)** Average values of the profiles displayed in E in the time windows identified in F, for all mice. **(H)** Same as C–G for the left pupil. The data and code underlying this figure is available in the following repository: https://doi.org/10.5281/zenodo.14635566.

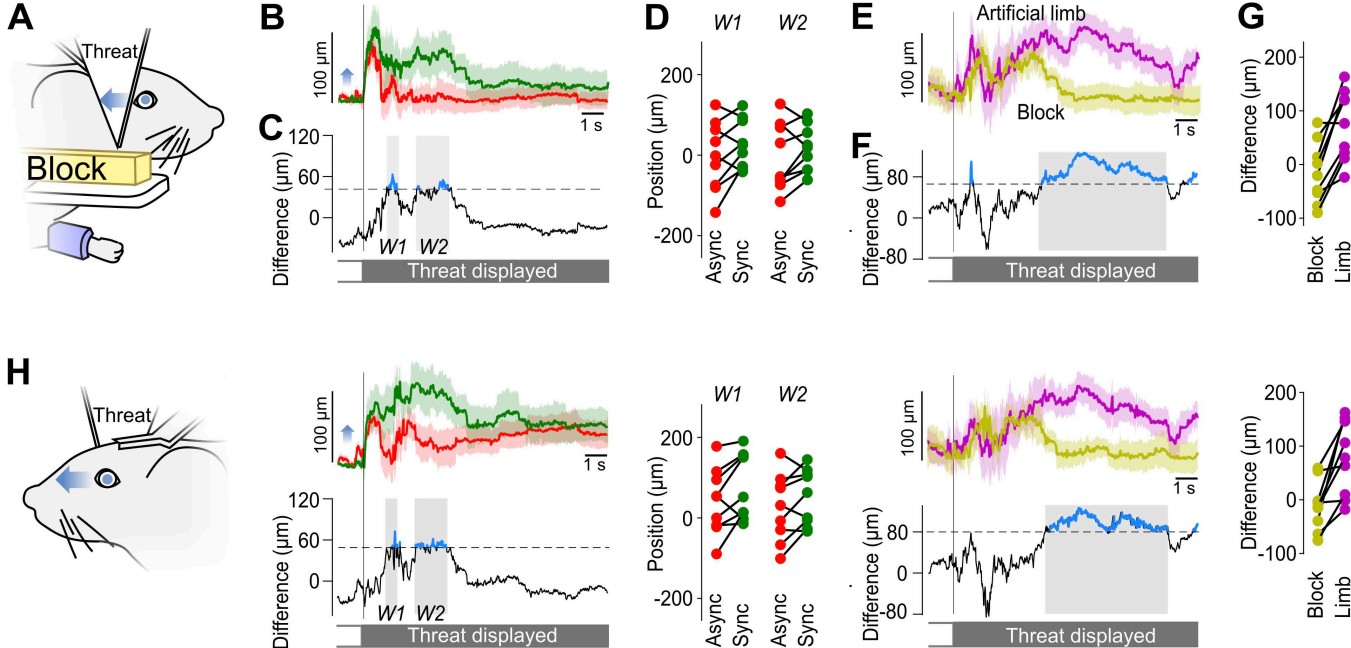

**Fig 3. Pupil shifts are shorter when the artificial limb is replaced by a white block.** (A) Schematic of the right side of the mouse during the white block experiment (depicted here in yellow for contrast enhancement of the figure). Blue arrow: general direction of the pupillary movement following the threat. (B) Average horizontal movements of the right pupil after the threat, normalized relative to the average position 1 s before the threat ($n = 9$ mice). Green line: synchronous pairing. Red line: asynchronous pairing. Light background: SEM. Blue arrow: direction of pupil movement as indicated in A. (C) Average difference between the right pupil movements in the two conditions in B. Blue sections: significant differences (Bootstrap-based test $p < 0.05$). Black dashed line: significance threshold. Gray background: W1 and W2 time windows selected for further quantification. (D) Average values of the profiles displayed in B in the time windows W1 and W2 for all mice. (E) Average time course of the difference in the right pupil response to the threat between synchronous and asynchronous pairings with the artificial limb (magenta line), and with the white block (yellow line). Light background: SEM. (F) Difference in sync/async contrast in threat response, as computed in E, between the artificial limb and the white block conditions. Blue sections: significant differences (Bootstrap-based test $p < 0.05$). Black dashed line: significance threshold. Gray background: time windows selected for further quantification. (G) Average values of the profiles displayed in E in the time window identified in F, for all mice. (H) Same as A–G for the left pupil. The data and code underlying this figure is available in the following repository: https://doi.org/10.5281/zenodo.14635566.

To further compare the reaction to the two shapes, we computed the difference in average right pupil shift between synchronous and asynchronous conditions across the two shapes (Fig 3E). We found that during a prolonged, late time window (Fig 3F), the threat reaction in the Artificial limb condition was significantly stronger than the reaction in the Block condition. The observations on the movements of the left eye pupil (Fig 3H) were fully consistent with these observations in the right pupil. To summarize, in these additional, "Block" experiments, we found limited signs of embodiment of a non-limb object. However, their strength was significantly smaller than in the case of an artificial forelimb.

## Threat response without pairing

Overall, the pupil movements in reaction to the threat after pairing in the Block condition were significantly smaller than after pairing in the Artificial limb condition. However, we found that even in the Block condition, there was a clear pupillary reaction to the threat, both following synchronous and asynchronous pairing (Fig 3B). We therefore asked what part of the pupillary reaction was solely due to a reflex pupillary reaction to the approaching threat, independent of the pairing of the mouse limb with either an artificial limb or a white block object. To measure this, we performed an additional experiment in which the same group of nine mice only received a threat, in the absence of any stimulation (Fig 4A). In this condition,

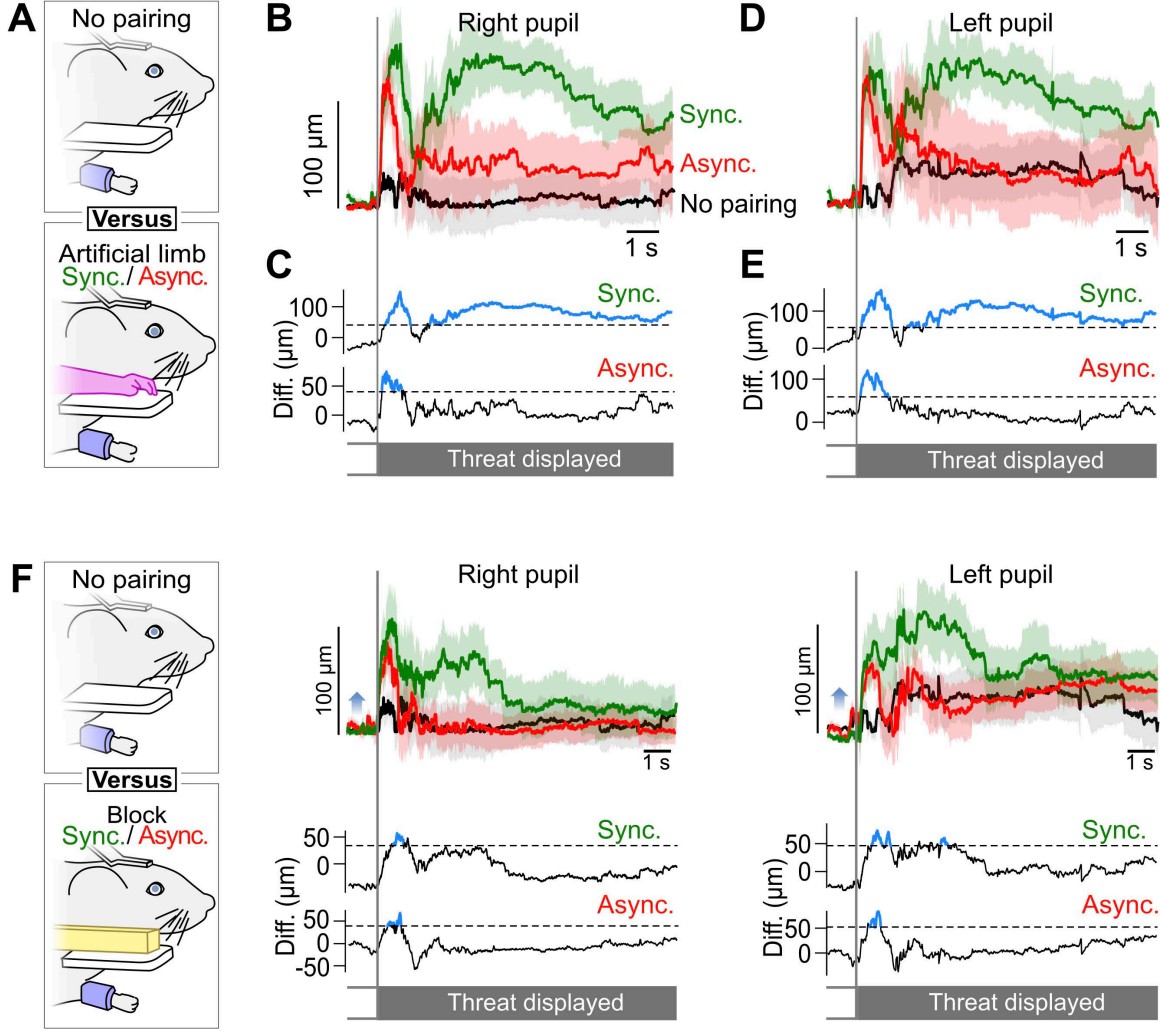

**Fig 4. Comparison of the threat reaction with vs. without pairing. (A)** Comparison of the eye movements in the No pairing vs. the Artificial limb conditions, with both synchronous and asynchronous pairing. In the No pairing condition, during 240 s, the mice were idle, and no artifact was presented to the mouse. After this waiting period, the threat was presented. **(B)** Average horizontal movements of the right pupil following the threat onset, normalized relative to the average position 1 s before the threat (*n* = 10). Light background: SEM. Black: pupil movements in the threat-only condition without pairing. Green and Red lines: Response to synchronous/asynchronous stimulations of the artificial limb, reproduced from Fig 2E. **(C)** Top: difference between the horizontal eye movements in response to synchronous pairing of the artificial limb, versus no-pairing. Blue sections: significant differences (Bootstrap-based test *p* < 0.05). Black dashed line: significance threshold. Bottom: asynchronous pairing versus no-pairing conditions. **(D)** Same as B for the left pupil horizontal movements. **(E)** Same as C for the left pupil horizontal movements. **(F)** Same as A to E for the No pairing versus the Block condition.

we found limited pupillary movements (black lines, Fig 4B, 4D). In particular, the rapid, onset phase of the reaction to the threat that we had noticed previously in all conditions almost disappeared. Compared with the synchronous pairing condition with the artificial limb, the pupil movements for left and right eyes were significantly smaller during the whole duration of the threat presentation (Fig 4B, 4D, black versus green lines; Fig 4C, 4E significance testing), and significantly smaller than in the asynchronous artificial limb pairing during the initial phase of the reaction (Fig 4A—4D, black versus red lines). They were even significantly smaller than in the Block condition, in particular during the onset phase of the pupil movements (Fig 4F).

We conclude from this additional experiment that the presentation of the threat by itself evokes only a mild reaction of the mouse, much smaller than when the threat comes after a visuo-tactile pairing. This suggests that pupil movements measured after pairings result from the perception of visual and tactile stimulations linked to the artifact. Notably, large pupil movements in the late phase of the response were only observed in the presence of the artificial limb, and only following the presentation of synchronous visual and tactile stimulations. Thus, we found a threat response only in the specific condition in which we could expect mice to embody the artificial limb, similarly to what happens in the human rubber hand experiment.

### Beyond pupil position

In addition to the pupil position, we also looked at the evolution of pupil diameter after the threat. We found a consistent increase in both right and left pupil diameters in all conditions after the arrival of the threat (S2 Fig). This increase happened both in the asynchronous and the synchronous pairings in the Artificial limb and the Block conditions. But was significantly smaller in the synchronous than in the asynchronous condition in the Artificial limb condition, while there was no significant difference in the Block condition.

Beyond eye tracking, most facial movements showed no significant embodiment-related variations. However, we did observe a consistent trend in whisking behavior (S2F Fig), and a significant difference in the speed of ear movement between the synchronous and asynchronous pairing with the artificial limb (S2G Fig).

These observations suggest that although pupil movements are the clearest indicator of embodiment in our experimental setting, full face tracking may consolidate this measurement and ensure more robust embodiment assessment in future experiments.

## Discussion

We have developed a protocol to probe the response of mice exposed to an analog of the "rubber-hand illusion" embodiment protocol. The mice forelimb—hidden from the mice sight—and an artificial forelimb positioned next to the mice were simultaneously brushed. Subsequently, we measured the response of mice to a threat to the artificial forelimb. We found that after a rapid excursion toward the threat, on average, the mice focused their gaze for several seconds toward the threat, while this was not the case in control experiments in which the stimulation of the mouse forelimb was not synchronous with the stimulation of the artificial limb.

### Position of the artificial limb with respect to the physiological limb

In our experiment, we exposed mice to a protocol that was directly derived from the design of the human rubber hand illusion. Our protocol was similar to the vertically stacked setups of the rubber hand illusion [7]. In our design, in order to adapt to the mouse anatomy, the artificial forelimb was placed on a platform located 1.2 cm on top of the mouse's real paw and 0.8 cm away horizontally. These distances were the minimum achievable shift to ensure that the artificial limb would be seen by the mouse. By this means, we attempted to maintain a congruent position of the artificial limb with respect to the real limb. Indeed, this positioning resulted in a total distance of less than 1.5 cm between the artificial and real limbs, which corresponds at the mouse scale to the 30 cm radius where the illusion can still be experienced in human experiments [27].

In the classical rubber hand illusion, participants are asked to remain static and to focus their attention on the rubber hand [5,28]. To be able to achieve this in mice, we head-fixed the animals and restrained the movements of the tested forelimb. This difference with the conditions in human experiments required habituation of the mice during 1–2 weeks before the actual pairing was performed. Note that instead of handcuffing, we initially attempted to condition the mice to hold their forelimb in place and to accept brush strokes on the forelimb without retracting it. However, we never achieved the repeatability and time stability of the forelimb posture that was required to induce the embodiment, then probe it.

## Brush stroke characteristics

Our mechanical brush stimulations occurred during 2 min at a frequency ranging between 0.6 and 2 Hz, which is consistent with the range of stimulations used in human experiments [29–31]. During individual brush stroke events, the brush swept the limb at around 34 mm/s. This falls in the range of stimulation speeds that are perceived as pleasant by humans [32] and close to ranges that have been used to generate pleasant touch in mice [33]. This may have enhanced the strength of the embodiment [34]. However, additional work to optimize the parameters of the stimulation may further enhance the pleasantness of the stimulation and the strength of the resulting embodiment.

## Artificial limb characteristics

In our experiments, we have asked if, in the mouse model, there was an impact of the artificial limb shape on the observed pupil shifts (Fig 3) after pairing. Human experiments have explored the limits of artificial limb embodiment by studying the impact of the visual similarity of an artificial hand to a real limb in the rubber hand illusion [15,17,18] and in virtual environments [35]. These experiments show higher embodiment scores for hand-shaped objects compared to non-hand-shaped objects. Zhang et al. showed that participants in virtual reality environments tend to look less at artificial hands with simple shapes as well [35]. To take into account these findings, we designed our artificial forelimb shape to be faithful to mouse forelimb anatomy [36], and we colored it to match the mouse fur and skin color using acrylic paint. To test for the specificity of the shape of the object being embodied, we replaced the artificial limb shape with a white rectangular block (Fig 3). We found that, in this condition, the differences in pupil shift between synchronous and asynchronous stimulations lasted for a shorter time. Although we did not find a complete reduction of the differences between the synchronous and asynchronous conditions, these results suggest that, consistent with human results, mice are more capable to embody a limb-like object rather than an arbitrary shape object.

## Readout of the embodiment of the artificial limb

Rubber hand illusion experiments in human subjects rely mostly on subjective questionnaires to assess the induced response. However, other non-verbal measures have been developed to quantify the strength of the responses. This includes the proprioceptive drift reported in the initial rubber-hand study [5], where participants are asked to close their eyes and use their free hand to localize the hand involved in the experiment. This reveals a shift of the perceived location toward the rubber limb after synchronous stimulations.

Another alternative to questionnaires is to probe reactions to a threat to the rubber hand. When the rubber limb is embodied in the test conditions, threatening it provokes an anxiety-like response that can be seen cortically as an increased activity in the insula and anterior cingulate cortex [10] as well as through SCR. These responses are also accompanied by participants' reports indicating an anticipation of pain, as well as facial, verbal, and motor signs of surprise or nervousness upon the threat or injury of the fake limb [9]. One such threat is the static exhibition of a sharp object, like a knife or a needle, next to the rubber hand [10]. However, in this case, the threatening aspect of the object stems from prior knowledge about sharp objects that may not be available to laboratory animals. These threats are therefore not relevant in animal experiments, and the previous study investigating this phenomenon in mice used the threat of a strong grasp of the tail by an operator [22]. However, we wanted our setup to be experimenter-independent. Therefore, we based our work on another threatening event: a fast-moving object approaching the artificial limb, which could be easily implemented with a stepper motor. We found this to be efficient in eliciting a clear facial reaction of the mice, while being similar to impact-based approaches in human paradigms that elicit SCR responses [37].

## The contribution of expectations to the rubber hand illusion strength

In recent years, there has been significant controversy surrounding the rubber hand illusion paradigm and the potential influence of participant expectations on the reported effects [38,39]. Studies suggest that participants' expectations and

hypnotic suggestibility could explain the subjective experience reported in this illusion, potentially confounding the experiment with a form of suggestibility or phenomenological control. However, the replication of rubber hand illusion-like effects in the mouse model—where complex cognitive expectations and bias are likely absent—provides compelling evidence that this illusion may not be solely driven by expectation. On the contrary, it suggests that multisensory integration processes indeed underlie the illusion, and that these processes may be shared across species.

## Pupil position is a key observable in mice

Our readout of the facial movements of the mice has revealed a clear contrast between the pupillary movements that were initiated by the threat following synchronous versus asynchronous stimulations. These findings are consistent with the observations in humans of a difference in the reaction to a menace in synchronous versus asynchronous pairing conditions [9,37]. Our videography of the mice revealed coupled right and left pupil shifts toward the artificial limb and threat, which lasted longer and were significantly more prominent in the synchronous pairing condition (Fig 2) and very low when only the threat was applied without pairing (Fig 4). In two previous studies that investigated tail embodiment in mice through an analog of the rubber hand illusion [22,23], head movements were reported as a reaction of the mice to grasping of the fake tail. However, in our experiments, the mice were head-fixed. We therefore hypothesize that the pupil movements that we observed were used by the mice to rotate their gaze despite the head fixation [40].

## Contributions to changes in pupil diameter

Beyond the position of the pupil, we also found significant differences in the dynamics of the pupil diameter (S2 Fig). After the threat of the artificial limb, we observed a dilation of the pupil, that was also present when only the threat was applied, without prior exposition to the stimulations. Pupil dilation has been shown to correlate with different arousal states [41,42], attention [43], as well as processing of startling stimuli [44] and fear conditioning [45]. We interpret this increase in pupil diameter as a sign that mice were strongly engaged by the arrival of the threat in all conditions in a similar manner.

On top of this overall trend, we noticed that the right pupil diameter was significantly larger in the asynchronous condition following the initial pupil dilation. This difference in pupil diameter did not match our expectations, as we had initially hypothesized that the incoming threat should trigger a stronger emotional reaction in the synchronous than in the asynchronous condition, and therefore a larger pupillary opening.

Several confounding factors that may explain this pupillary behavior. First, in the synchronous condition, the mice focused their gaze toward the bright white plastic threat more than in the asynchronous condition. They may therefore have been exposed to a higher level of illumination, that they would have compensated by reducing their pupillary diameter. Second, the dynamics of pupil size are known to reflect cognitive processes, including memorization. For instance, studies have shown that pupil constriction is stronger when individuals are exposed to images that they later recall [46] or when encountering novel stimuli [47]. Finally, pupil size adjustments are linked to the rapid switching between rod-driven and cone-driven vision systems, which allows animals to adapt their visual perception to specific environmental cues [48,49]. These insights may help in explaining the delayed reduction in pupil size that we observed after the initial dilation across all conditions, including the Artificial limb and Block conditions.

## Toward the tracking of facial mimics for embodiment

Beyond individual features of the mouse face, such as the position or diameter of the pupil, the tracking of the overall facial expression has emerged as a relevant strategy to probe different emotional states across several species, and in particular in rodents. When provoking a negative reaction such as fear, studies show that the animals' grimace differs significantly from baseline [50,51], in particular by modulating the ear and whisker pad areas of the face. We therefore tracked changes in the speed of the ear and vibrissae movements. We did find that the ear and whisker pad moved faster in response to the threat of the limb-like object after synchronous stimulations compared to the asynchronous control,

although these effects were less prominent than the ones observed for the pupils (S2 Fig). However, in our analysis, we failed to identify coherent clusters of coherent change across the face of the mice, and thereby focused our analysis on separate points of the face—and in particular the pupil position—rather than on multidimensional face features.

Using the high-speed imaging of a large part of the mouse body during the embodiment experiment, we could also search for signs of embodiment beyond the mouse face. In particular, we asked if the hand-cuffed forelimb displayed retraction movements in reaction to the threat. However, the paw being securely restricted, the attempted movements may not have been visible enough to be captured. Overall, we found that embodiment-related movements could only be detected on the face of the mice.

### The challenge of response variability

We have found a large variability in the eye movement response to the threat during multiple trials performed by the same mouse (Fig 2A, right). This variability across trials, that can be observed in human experiments [52] suggests that the threat response of the mice is not only shaped by limb embodiment, but also by additional factors, including the general level of attention of the mice, as well as the visual focus of the mouse at the onset of the threat. Further, similar to human variability across subjects [52], we have found that some mice can show strong signs of embodiment, while other responded on average in the same way to the synchronous and asynchronous conditions (Fig 2G).

Another source of variability is the potential habituation of the mice to the threat that we used to probe the illusion strength [53,54]. To minimize this issue, we presented the limb threat to the mice only twice a day during 5 days (synchronous and asynchronous conditions). We then waited 40 days before presenting a new set of conditions to the same mice (S1 Fig). Despite this limited exposition to the forelimb threat, we expect that part of the smaller response in the Block experiment (Fig 3) may be attributable to this threat habituation, although it is unlikely that it could be key to the significantly smaller synchronous/asynchronous difference in gaze shift in the Block experiment when compared to the Limb experiment.

This variability will need to be tackled upon exploring the brain circuitry that supports forelimb embodiment, both by focusing on mice that display a stronger sensitivity to artificial stimulations, and by optimizing the sensory inputs and the menace that are presented to the subject.

### Cognition in the mouse model

Our work has practical implications for the study of forelimb embodiment in a model that offers unparalleled experimental venues. However, beyond this, it also supports the idea [22] that rodents can display behavioral correlates of embodiment in settings that are known to trigger embodiment in humans. This adds to a series of recent findings that suggest that rodents, and mice in particular, can display some capabilities that have been associated with higher cognitive functions in the past. In particular, there has been recent evidence [55] that mice respond to the mirror task in ways that are consistent with several features of self-recognition in humans [56]. Other lines of research have shown, first in rats [57,58], and then in mice [59] that rodents respond to tickling by emitting ultrasound "laughter", and that they display playfulness in game contexts. Taken together, these observations suggest that mice are a relevant model to explore the cortical circuitry that underlies the cognitive functions related to self-recognition and social interactions.

## Materials and methods

### Ethical compliance and mice

All animal experiments were performed according to European and French law as well as CNRS guidelines and were approved by the French Ministry for Research (Ethical Committee 59 "Comité d'éthique en matière d'expérimentation animale Paris Centre et Sud", authorization 25932-2020060813556163v7). In order to reduce the number of mice involved in research experiments, we carried our experiment on EMX-Cre mice (Jax #005628) that were raised toward

the maintenance of transgenic lines in the institute animal house, but were not directly used in other experiments. We could take advantage of these mice as their genotype is normal, and we found no noticeable difference in behavior when compared to wild-type C57BL/6 mice. Mice were housed in cages of 5 in a non-inverted light cycle. They had access to an enriched environment (a wheel, wooden and plastic toys, as well as nesting material). We used 10 mice (5 males, 5 females) that were aged between 60 and 100 days at the time of the experiment.

### Surgical procedure

During the experiments, the mice were head-fixed. This allowed us to maintain the mouse and its paw immobile to ensure the stability of the tactile stimulations during the experiment. It also allowed for a direct translation of our protocol into the many head-fixed experimental setups for brain imaging and optogenetic stimulation. Implantation surgeries were carried under Isoflurane anesthesia (4% for induction and 1%–1.5% for maintenance). Surgeries were performed on a heated pad, while the mouse was held by a nose clamp. After a subcutaneous injection of lidocaine (4 mg/kg) the scalp was removed, the conjunctive tissue resected, and the skull was cleaned. A metallic head-fixation plate was then bonded to the skull using a cyanoacrylate glue primer topped with dental cement. Finally, the mice received a subcutaneous injection of anti-inflammatory medication (Meloxicam, 1–8 mg/kg) and were monitored during their recovery in a temperature-regulated cage.

### Recovery and habituation

The mice were placed in the experimental setup for the first time after a 5-day recovery phase in their home cage. During this habituation phase, the mice were head-restrained, with their body placed in a pod. They were given water with sugar to associate habituation with a positive reward. After an initial 10-min session with head-fixation only, the next four head-fixation sessions lasted 20 min. They were coupled with movement restriction of the right forelimb. The paw was restricted using a custom-made handcuff that was adjusted and secured on a dedicated, foam cushioned location on the pod (Fig 1A).

### Design of the artificial paw illusion

A 3D model of a right mouse forelimb was designed, based on a 3D atlas of adult C57BL/6 mice derived from micro-CT sections [36]. It was printed using a resin 3D printer (FormLabs Form3B, Gray Flexible Resin) and painted using acrylic paint to match the black color of the fur of C57BL/6 mice. It was placed aside the head-fixed mice, and illuminated by a ray of visible, white light (while the rest of the setup was only illuminated by infrared lighting for imaging). Meanwhile, the actual right paw of the mouse was restricted and hidden below the platform holding the artificial limb.

After a 120 s waiting period, the mouse was exposed during 120 s to a series of soft strokes on the real and artificial paw, applied by paint brushes mounted on servo motors (Make Block Smart Servo). One "brush stroke" event was achieved by the brush making a back-and-forth movement on a 6 mm distance on the paw that lasts for 300 ms between touch onset and offset (Fig 1B). During the pairing time, the brush strokes were applied both on the hairy side of the mouse right forelimb and on the artificial limb at random, Poisson-distributed intervals ranging from 600 to 2000 ms. In the synchronous condition, the two brushes applied the stimulation at the same time, while in the asynchronous condition, each brush was activated at a different, randomized interval, such that the visual input from the artificial limb did not match the tactile input applied to the real forelimb (Fig 1B).

Finally, 240 s after the beginning of the trial and right after the end of the brush stimulations, a threat to the fake forelimb was presented to the mouse (Fig 1A, right). This was achieved by using a stepper motor (17HS15-0404S, OSM Technology) to rapidly rotate by 180° an arrowhead-like white plastic object, moving it from a hidden position to a position next to the fake limb, at a speed of 0.8 m/s. The threat stopped less than 1 cm away from the artificial limb and stayed at that position for 10 s (Fig 1A, 1C).

 

## Brush stroke pairing conditions

The embodiment experiments using the artificial limb lasted 1 week. We ran five sessions for each experiment (1 session per day). During each session, the mice were exposed to two trials (synchronous and asynchronous) whose order changed at each session. We chose this multi-day design to minimize mouse fatigue and reduce the effect of the habituation curve that would have been potentially more prominent if all the trials were run on the same day.

Forty days later, we performed an additional control experiment. The goal of this experiment was first to probe the baseline reaction of the mice to the threat alone. Second, we wanted to explore if the mice would react to the threat of any object stimulated in synchrony with their paw, or if this was limited to artificial limbs resembling their own. Mouse 20 was not part of this experiment as the animal had to be removed from the experimental pool for veterinary reasons. The third trial for mouse 19 was also removed due to faulty operation of the high-speed camera during this session. This control experiment was otherwise identical to the initial series of experiments, but this time we exposed the animals to the threat alone followed by two trials (synchronous and asynchronous) where the fake paw was replaced by a white plastic block of the same size as the artificial limb (see S1 Fig for trial order).

## Face imaging and tracking

The mice were imaged at 200 Hz with two 1440×1080 px monochrome cameras capturing the right and left facial expressions (Fig 1C) using a custom imaging system (RD Vision, France). The reactions of the mice to the stimulations and the threat were recorded, and the videos analyzed with DeepLabCut version 2.3 [60]. We trained two networks (one network for each side of the mouse's face) on 120 labeled images of 10 different mice to track a series of points of interest on the animals face (blue dots in Fig 1C), including the center of the pupil position, two points of the pupil for the diameter, ear, and either the C1 or the B1 whisker (depending on which one was more visible on camera).

## Statistical analysis

To correct for baseline shifts, we subtracted the mean position of the tracked position measured during the 120 s baseline that proceeded the brush stimulations. When looking at the effect after the threat on pupil shifts and diameter, we subtracted the mean values 1 s before the threat so we could normalize to pre-threat positions that may not be the same from trial to trial.

To detect significant differences of the trajectory of the pupil between the synchronous and asynchronous conditions, we built a bootstrap-based analysis. We focused our detection of significant deviations on the time window ranging from 1 s before the threat, to 10 s after. To estimate the significance of the differences over this 11 s period, we first computed a distribution of the maximal deviations observed in 11 s segments of control data. To build this distribution, we took advantage of the 120 s of baseline that we systematically acquired before the pairing step of the experiments. For each of the 10 mice, we computed the difference between the baseline position of the pupil recorded just before the synchronous versus asynchronous conditions. We then repeated 10,000 times the following calculation: we pulled an 11 s segment from the 120 s baseline difference at a random time for each mouse (different for each iteration and mouse), and computed the average across mice, then identified the maximum of this 11 s time series. By this means, we obtained a 10,000-samples distribution of the maximum difference between synchronous and asynchronous conditions that can be expected to occur during 11 s of baseline in our dataset. Finally, we extracted from this distribution the 5% false positive threshold. This threshold was recomputed for each of the experimental conditions that we explored in this manuscript (Artificial limb, Block, No pairing). Note that in the significance analysis, there is no subtraction of the baseline, leading to non-zero synchronous/asynchronous difference at the start of the threat (for instance in Fig 3C) while the baseline is subtracted for the raw pupil positions (for instance in Fig 3B), in order to show the threat response amplitude.

When comparing the strength of the sync/async difference in the Artificial Limb versus the Block experiments, we used the same bootstrap strategy on this difference of differences to obtain an adequate significance threshold.

All analysis code was built in a Python environment. The corresponding Jupyter Notebooks and required data are archived: https://doi.org/10.5281/zenodo.14635566

## Supporting information

**S1 Fig. Trial order for experimental protocol. (A)** Listing of all experimental conditions. **(B)** Order of the experimental conditions during the first experiment, in which ten EMX-Cre C57/BL6 mice received both synchronous and asynchronous stimulations on the artificial limb. **(C)** Second experiment, in which nine of the ten mice from B were tested on three additional, control protocols, 40 days after the first experiment.
(TIFF)

**S2 Fig. Behavioral markers of embodiment beyond pupil position. (A)** Pupils are less dilated in response to the threat in the synchronous condition (green line) than in the asynchronous condition (red line). Average vertical diameter of the right pupil after the threat in the artificial limb condition, normalized relative to the mean position in the second before the threat ($n = 10$). Light background: SEM. **(B)** Average difference between the right pupil diameter between the two conditions in A. Blue sections: significant differences (Bootstrap based test $p < 0.05$). Black dashed line: significance threshold. Light gray background: time window selected for further investigation. **(C)** Average difference in the time window identified in B, for each individual mouse, in the synchronous versus asynchronous condition. **(D)** Difference between the sync/async contrasts observed in the artificial limb versus Block conditions. Blue sections: significant differences (Bootstrap based test $p < 0.05$). Black dashed line: significance threshold. **(E)** Same as A–D for the left pupil diameter. **(F)** Same as A, B, D for the absolute value of the instantaneous speed of the left ear. The average left ear speed following a threat increased more in synchronous versus asynchronous pairings with the artificial limb. This difference crossed at several time points the significance threshold. We did not find a significant ear movement difference between the threat response in the conditions of Artificial limb versus Block. **(G)** Same as F for the speed of either the B1 or C1 left whisker. The data and code underlying this figure is available in the following repository: https://doi.org/10.5281/zenodo.14635566.
(TIFF)

## Acknowledgments

We thank Isabelle Ferezou for advice and support throughout the project. We thank Amaury François, Frédérique de Vignemont, and Matthew Larkum for insightful exchanges on an earlier version of the manuscript.

## Author contributions

**Conceptualization:** Zineb Hayatou, Antoine Chaillet, Daniel E Shulz, Valérie Ego-Stengel, Luc Estebanez.

**Data curation:** Luc Estebanez.

**Formal analysis:** Luc Estebanez.

**Funding acquisition:** Antoine Chaillet, Daniel E Shulz, Valérie Ego-Stengel, Luc Estebanez.

**Investigation:** Zineb Hayatou.

**Methodology:** Zineb Hayatou.

**Project administration:** Luc Estebanez.

**Resources:** Hongkai Wang, Luc Estebanez.

**Supervision:** Valérie Ego-Stengel, Luc Estebanez.

**Validation:** Luc Estebanez.

**Writing – original draft:** Zineb Hayatou, Luc Estebanez.

**Writing – review & editing:** Zineb Hayatou, Daniel E Shulz, Luc Estebanez.

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
