## [Editor Report · Decision Letter 0]

10 Jan 2025

Dear Dr estebanez, 

Thank you for submitting your manuscript entitled "Embodiment of an artificial limb in the mouse model" for consideration as a Research Article by PLOS Biology.

Your manuscript has now been evaluated by the PLOS Biology editorial staff as well as by an academic editor with relevant expertise and I am writing to let you know that we would like to send your submission out for external peer review.

Once your full submission is complete, your paper will undergo a series of checks in preparation for peer review. After your manuscript has passed the checks it will be sent out for review. To provide the metadata for your submission, please Login to Editorial Manager (https://www.editorialmanager.com/pbiology) within two working days, i.e. by Jan 12 2025 11:59PM.

Kind regards,

Christian

Christian Schnell, PhD

Senior Editor

PLOS Biology

cschnell@plos.org

---

## [Decision Letter · Decision Letter 1]

7 Mar 2025

Dear Dr Estebanez,

Thank you for your patience while your manuscript "Embodiment of an artificial limb in the mouse model" went through peer-review at PLOS Biology. Your manuscript has now been evaluated by the PLOS Biology editors, an Academic Editor with relevant expertise, and by several independent reviewers.

In light of the reviews, which you will find at the end of this email, we are pleased to offer you the opportunity to address the comments from the reviewers in a revision that we anticipate should not take you very long. We will then assess your revised manuscript and your response to the reviewers' comments with our Academic Editor aiming to avoid further rounds of peer-review, although we might need to consult with the reviewers, depending on the nature of the revisions.

Please note, as mentioned in our previous email, we think that your manuscript would be best suited for our ***Short Reports*** format. Please select this category when resubmitting your revised manuscript. There is a limitation of four figures for a Short Report, but no limitations in terms of word count or references.

**IMPORTANT - SUBMITTING YOUR REVISION**

*Resubmission Checklist*

*Published Peer Review*

*PLOS Data Policy*

*Blot and Gel Data Policy*

Sincerely,

Christian

Christian Schnell, PhD

Senior Editor

PLOS Biology

cschnell@plos.org

REVIEWS:

Reviewer #1: Dear Editors and authors,

In a new study Hayatou et al. aimed to establish a rodent model of the widely used "rubber hand illusion" paradigm in humans. This is an important task. While previous studies using the human paradigm allowed studying brain activity correlated to embodiment of artificial objects and thereby provided insight into the brain-wide organization, the neural- and circuit mechanisms underlying embodiment could not be investigated so far. This is mainly due to a lack of a laboratory animal model. The authors therefore elegantly adapted the human paradigm to rodents by using certain facial expression parameters as a readout for embodiment. In particular, they show that mice adjust their eyes gazing at the moving threat stimulus and that this adjustment is most pronounced if mice were previously brushed synchronously at their real and their artificial forepaw. If the brushing was performed non-synchronously the authors observed less gaze adjustments and if no brushing was preceded the threat stimulus the gaze adjustment was almost absent. The authors also performed proper control experiments. E.g. they showed that an object, which did not resemble details of the real forepaw generated less gaze adjustments in response to a threat stimulus, suggesting that that object was not embodied as strongly as the more realistic artificial limb.

Taken together, this study provides an important methodological advancement, which may be used in the future to probe brain functions related to body coordination and perception and possibly to address mechanistic questions regarding clinical conditions such as phantom limb pain. I strongly support publication of this work in PLOS Biology, however, the authors need to address some minor issues that may improve the quality of the manuscript.

Minor: 

-At the end of the introduction the authors state: "Consistent with the human rubber hand illusion, the mice focused significantly more…". Here the authors should provide citations that show this gaze effect in humans.

-The authors show 3 different experiments all carried out with the same mice. While this approach has the advantage that one gets within-animal data it has the disadvantage that possible habituation processes cannot be excluded. The authors should address this in the discussion.

-In Figure 4 it does not become evident why in panel F (top) there is no depiction of analogous traces shown in panel A (top, green, red and black traces). The authors should have this data, so why is it not shown?

-In the discussion the authors provide valid arguments why the head and the forepaw were fixed. It may, however, still make sense to analyze the restrained forepaw/limb and see whether certain twitches (potentially of the fingers?) may also correlate with the degree of embodiment. Also, one may think of training mice to leave their forepaw at a desired place, which would make the paradigm more complicated but also more similar to the human paradigm. It would be good if the authors discuss this issue.

-The authors also discuss the non-intuitive results of their supplementary figure 2. There they show that pupil diameter was bigger in the asynchronous condition compared with the synchronous one. The authors state that cognitive processes may be involved in pupil diameter control but how this explains the observed results remains elusive. As far as I understand these results are difficult to match with existing findings from human studies as the degree of embodiment is usually positively correlated with levels of arousal and not negatively (as suggested by the results in supplementary figure 2). This issue should be discussed more intensively.

-In the methods the authors should state at which angle the arrowhead-like white plastic was moved (in horizontal and vertical dimension).

Reviewer #2: This study describes what could become a useful resource for trying to understand circuitry underlying limb embodiment and how it operates in the mammalian brain. Namely, it introduces an assay which replicates in mice similar behaviors to those found in the 'rubber hand illusion' (RHI) in humans. 

My sense of this manuscript is that it is more a novel resource than a report of novel findings and insights. It seems to me that the value provided by replicating the RHI in mice would be to allow investigation of its neural circuit basis in a tractable model, which the manuscript lays the ground for, while stopping short of providing any neural evidence. 

More specifically: as the authors say in the Discussion, there has been controversy over whether human RHI arises purely from bottom-up multimodal integration leading to embodiment, or whether it is affected by top-down phenomena such as expectation and hypnotizability (Lush et al, 2020; Thériault et al, 2022). The current manuscript provides evidence suggesting that something similar to the human phenomenon is generated in mice and probably emerges without the need for hypnotizability, therefore providing a useful resource/experimental model to the community. This might lead to mechanistic accounts of RHI emergence through multimodal integration in mice, which could be highly interesting and relevant. This manuscript does set up an appropriate behavioral design but I don't think its novel findings are substantive enough for publication as a PLOS Biology research article. 

Are data values properly aligned to axes in all figure panels? In Fig. 3B, the green and red curves seem to start at about the same level, and the green curve then stays above the red curve. Yet in Fig. 3C the difference value starts out highly negative, and then becomes negative again about a second after the end of W2, when green is still clearly greater than red in panel 3B. Please check alignment to axis values of these and other plots. 

Reviewer #3: This paper is an elegant and novel contribution to the literature. The authors perform ingenious, well designed experiments to develop a paradigm for the study of limb embodiment in mice. With the current arsenal of tools for dissecting neural circuitry in this model, this could represent a panoply of new questions.

This reviewer could not find fault with the science; the only nagging big picture question was "how much does our inability to communicate with mice limit what can be done here?" The authors address this in the discussion, especially from the perspective of what can be correlated with various cognitive and physiological states (eye movements, whisking). So it's more of a curiousity that criticism.

There were a couple tiny errors, otherwise the manuscript is well written and presented and publishable.

INTRO:

This sentence not grammatical:

Further, the lack

of embodiment of prosthetic substitutes participates in the sensations that are perceived as arising from

the "phantom" of the missing limb, including painful perceptions4.

discussion

typo - laugher -> laughter

---

## [Decision Letter · Decision Letter 2]

24 Apr 2025

Dear Dr Estebanez,

Thank you for your patience while we considered your revised manuscript "Embodiment of an artificial limb in the mouse model" for publication as a Research Article at PLOS Biology. This revised version of your manuscript has been evaluated by the PLOS Biology editors, the Academic Editor and one of the original reviewers.

Based on the reviews and on our Academic Editor's assessment of your revision, we are likely to accept this manuscript for publication as a Short Report, provided you satisfactorily address the following data and other policy-related requests:

* We would like to suggest a slightly different title to enhance readability: "Embodiment of an artificial limb in mice"

* Please add the links to the funding agencies in the Financial Disclosure statement in the manuscript details.

* Please change the Article type to "Short Reports" during the resubmission of your revised manuscript.

* DATA POLICY:

Regardless of the method selected, please ensure that you provide the individual numerical values that underlie the summary data displayed in the following figure panels as they are essential for readers to assess your analysis and to reproduce it: 2G, 3DG and S2CE.

* CODE POLICY

We expect to receive your revised manuscript within two weeks. 

*Published Peer Review History*

*Press*

Sincerely,

Christian

Christian Schnell, PhD

Senior Editor

cschnell@plos.org

PLOS Biology

Reviewer remarks:

Reviewer #1 (Eduard Maier): The authors addressed all my points. I support publication in PLOS Biology

---

## [Editor Report · Decision Letter 3]

30 Apr 2025

Dear Dr Estebanez,

Thank you for the submission of your revised Short Reports "Embodiment of an artificial limb in mice" for publication in PLOS Biology. On behalf of my colleagues and the Academic Editor, Ann Clemens, I am pleased to say that we can in principle accept your manuscript for publication, provided you address any remaining formatting and reporting issues. These will be detailed in an email you should receive within 2-3 business days from our colleagues in the journal operations team; no action is required from you until then. Please note that we will not be able to formally accept your manuscript and schedule it for publication until you have completed any requested changes.

When you attend to the requests to come, please also add a reference to the source data in the legends of Figure 3 and S2. They are currently only referenced in the legend of Figure 2. 

PRESS

Sincerely, 

Christian

Christian Schnell, PhD

Senior Editor

PLOS Biology

cschnell@plos.org